# Association between Early Life Famine Exposure and Metabolic Syndrome in Adulthood

**DOI:** 10.3390/nu14142881

**Published:** 2022-07-13

**Authors:** Fan Yao, Liyun Zhao, Yuxiang Yang, Wei Piao, Hongyun Fang, Lahong Ju, Qiya Guo, Dongmei Yu

**Affiliations:** Key Laboratory of Trace Element Nutrition of National Health Commission, National Institute for Nutrition and Health, Chinese Center for Disease Control and Prevention, Beijing 100050, China; yao_f@foxmail.com (F.Y.); zhaoly@ninh.chinacdc.cn (L.Z.); yxyang_ninhccdc@126.com (Y.Y.); piaowei@ninh.chinacdc.cn (W.P.); fanghy@ninh.chinacdc.cn (H.F.); julh@ninh.chinacdc.cn (L.J.); guo_cecilia@foxmail.com (Q.G.)

**Keywords:** famine exposure, childhood, fetal life, metabolic syndrome

## Abstract

To analyze the relationship between famine exposure at different stages of early life and the risk of developing metabolic syndrome (MetS) in adulthood, 11,865 study participants from the 2015 Chinese Adult Chronic Disease and Nutrition Surveillance Program were enrolled and then divided into a non-exposed group, fetal exposure group, early childhood exposure group, middle childhood exposure group and late childhood exposure group according to their birth time and famine exposure. MetS was defined by the NCEP ATP III criteria. Using logistic regression to explore the association between famine exposure at different stages of early life and the increased risk of developing MetS in adulthood. After adjusting other factors, compared with the non-exposure group, famine exposure during the fetal period (OR = 1.23, 95% CI = 1.00–1.51), early childhood (OR = 1.44, 95% CI = 1.11–1.87), middle childhood (OR = 1.50, 95% CI = 1.13–1.99) and late childhood (OR = 1.67, 95% CI = 1.21–2.30) had a higher risk of developing MetS in adulthood. Stratified analysis found an association between early life famine exposure and the increased risk of MetS in adulthood in females, overweight or obese participants and those who lived in areas of severe famine, in city areas and in southern China. Compared with the non-exposed group, the fetal, early childhood, middle childhood and late childhood famine-exposed groups are more likely to suffer from MetS in adulthood, especially the subjects who are females, overweight or obese and had lived in severe famine areas, city areas and southern China.

## 1. Introduction

Metabolic syndrome (MetS) is a group of clinical syndromes characterized by the aggregation of multiple disease states such as abdominal obesity, hypertension, dyslipidemia, abnormal glucose metabolism and insulin resistance in individuals [1]. In recent years, its prevalence has been on the rise globally. From 2003 to 2012, the prevalence of MetS in the US increased from 32.9% to 34.7% [2]. In Korea, the age-adjusted prevalence of MetS increased from 24.9% in 1998 to 31.3% in 2007 [3]. In 2001, the results showed that in 31 Chinese provinces, the age-adjusted prevalence of MetS in China was 13.7% (9.8% for men and 17.8% for women) [4]. From 2010 to 2012, its prevalence among Chinese adults had increased to 24.2% (24.6% for men and 23.8% for women) [5]. Age, education level, smoking and family history of diabetes are risk factors for MetS [6]. Other risk factors need to be explored further.

In the 1980s, Professor Barker in the UK found that intrauterine fetal dysplasia was strongly associated with cardiovascular disease in adulthood [7] and accordingly suggested the hypothesis of fetal origin of adult disease [8]: when maternal malnutrition occurs, the fetus redistributes the limited nutrients to prioritize the development of vital organs such as the brain for survival, while others such as pancreas and liver undergo physiological and metabolic programmatic changes that evolve into disease in adulthood. Previously, many animal model studies have supported this hypothesis [9,10]. For ethical reasons, there are no relevant studies to test the hypothesis in humans. However, famines that have occurred throughout human history provide a quasi-experimental context to verify the hypothesis that the early life famine has negative effects in adulthood, afterwards providing more direct evidence on this field in humans. Based on the data from the 2015 Chinese Adult Chronic Disease and Nutrition Surveillance (CACDNS 2015), the current study analyzes the association between different famine exposure stages in early life and MetS in adulthood among Chinese populations.

## 2. Materials and Methods

### 2.1. Data Source

The data were collected from CACDNS 2015, which adopted a stratified, multistage, random sampling method to recruit representative participants from 31 provinces/municipalities/autonomous regions of China. Detailed information about this survey is given in our previous report [11]. The protocol for the current study was approved by the Ethics Committee of the National Institute for Nutrition and Health, and the Chinese Centre for Disease Control and Prevention (approval number: 201519-A). All the participants signed an informed consent form prior to joining the study.

### 2.2. Research Subjects

Among adults aged over 18 years in the CACDNS 2015, those born between 01.10.1952 and 30.09.1964 were selected first. However, the famine in China spanned a long period at that time and the onset and end of the famine were inconsistent across provinces and regions. To minimize misclassification, participants born between the period of 01.10.1958 to 30.09.1959 and 01.10.1961 to 30.09.1962 were excluded by referring to the inclusion and exclusion criteria of previous literature [12]. Furthermore, the study population was divided into five groups according to the time of birth and famine exposure:(1)Non-exposed group: adults born between 01.10.1962 and 30.09.1964.(2)Fetal exposure group: adults born between 01.10.1959 and 30.09.1961.(3)Early childhood exposure group: adults born between 01.10.1956 and 30.09.1958.(4)Middle childhood exposure group: adults born between 01.10.1954 and 30.09.1956.(5)Late childhood exposure group: adults born between 01.10.1952 and 30.09.1954.

### 2.3. Survey Content and Method

The 2015 Chinese Adult Chronic Disease and Nutrition Surveillance was conducted in four parts: inquiry survey, medical physical examination, dietary survey and laboratory testing. This study incorporates basic household and personal information from the inquiry survey; height, weight, waist circumference and blood pressure from the medical physical examination; 24 h dietary retrospective survey data from the dietary survey; and blood lipids and blood glucose from the laboratory tests.

### 2.4. Quality Control

To ensure the quality of monitoring, the national working group of CACDNS 2015 developed monitoring quality control programs and supervised their implementation. During the monitoring process, the program, manual and questionnaire; training and assessment; equipment and reagents; and data entry and cleaning were all unified and implemented.

### 2.5. MetS Diagnostic Criteria and Its Related Definitions

#### 2.5.1. Diagnostic Criteria

MetS was defined according to the National Cholesterol Education Programme Adult Treatment Panel III (NCEP-ATP III) [13] as having three or more of the following factors:(1)A waist circumference of ≥90 cm for men and ≥80 cm for women;(2)A systolic blood pressure of ≥130 mmHg or a diastolic blood pressure of ≥85 mmHg or receiving anti-hypertension treatment;(3)A fasting triglyceride level of ≥1.7 mmol/L or receiving corresponding treatment;(4)A high-density lipoprotein cholesterol (HDL-C) level of <1.03 mmol/L for men and <1.30 mmol/L for women or receiving corresponding treatment;(5)A fasting plasma glucose (FPG) level of ≥5.6 mmol/L or receiving anti-diabetes treatment or reporting previously physician-diagnosed diabetes.

#### 2.5.2. Definition of Famine Severity

50% of excess mortality rate was used as a cut-off value to distinguish the severity of famine, in accordance with the methodology used in previous studies [14]. Excess mortality rates ≥50% were defined as areas severely affected by famine and excess mortality rates <50% were defined as areas less severely affected by famine.

#### 2.5.3. Relevant Definitions

(1)Area of the country: the southern provinces are Shanghai, Jiangsu, Zhejiang, Anhui, Fujian, Jiangxi, Hubei, Hunan, Guangdong, Guangxi, Hainan, Chongqing, Sichuan, Guizhou and Yunnan, 15 provinces in total. The northern provinces are Beijing, Tianjin, Hebei, Shanxi, Inner Mongolia, Liaoning, Jilin, Heilongjiang, Shandong, Henan, Tibet, Shaanxi, Gansu, Ningxia, Qinghai and Xinjiang, 16 provinces in total,(2)Body mass index (BMI): described by non-overweight or obese group (BMI < 24 kg/m^2^) and overweight or obese group (BMI ≥ 24 kg/m^2^).(3)Education level: low (primary school or below), medium (junior school) and high (high school and higher).(4)Income: divided into low level, medium level, high level and very high level by quintile.(5)Smoking was divided into smoking and non-smoking.(6)Drinking was divided into drinking and non-drinking.(7)Physically inactive: within one week, less than 150 min of moderate-intensity activity, less than 75 min of high-intensity activity or less than 150 min of above activity [15].(8)Family history of hypertension (or diabetes) was defined as one or more lineal relatives being diagnosed with hypertension (or diabetes).(9)Dietary pattern: The dietary pattern was established by using factor analysis, which was divided into egg, milk and fruit pattern, aquatic vegetable and meat pattern and staple food soybean and nut pattern.

### 2.6. Statistical Analysis

#### 2.6.1. Descriptions and Tests

SAS 9.4 software (SAS Institute Inc., Cary, NC, USA) was used for all the process of statistical analysis in current study. Statistical differences were considered at *p* < 0.05. Non-normally distributed measures, expressed as M(IQR), were compared using the Kruskal–Wallis rank sum test and comparisons between each group were achieved using a two-by-two comparison method. Count data were described by *n* (%), and chi-square test was used for comparison. Prevalence of MetS was calculated by complex sampling using data from the sixth census of the National Bureau of Statistics of China and comparisons were made using the chi-square test with Rao-Scott modified weights.

#### 2.6.2. Association between Famine Exposure and MetS in Adulthood

Unconditional logistic regression model was applied in this part. Because of the effect of age on the prevalence of MetS, the mean age of the fetal and childhood exposure groups was greater than that of the non-exposure group. Thus, to adjust the effect of age on the risk of MetS, age was included in the model as a categorical variable. Model I was adjusted for sex and age; Model II was further adjusted for the area of the country, location of residence, education, income, smoking, drinking, physical activity, dietary pattern, family history of hypertension and family history of diabetes. Model III was further adjusted for BMI and famine severity. To explore the association between early life famine exposure and risk of MetS in adulthood across sex, BMI, famine severity, residential location, and area of the country, stratification was performed by sex, BMI, famine severity, residential location, and area of the country.

## 3. Results

### 3.1. Sample Description

A total of 11,865 study subjects with a median age of 58.4 years were included. In total, 5306 (44.7%) were male and 6559 (55.3%) were female; 6134 (51.7%) lived in the south and 5731 (48.3%) in the north; and 5153 (43.4%) in a city and 6712 (56.6%) in the countryside. The sample size of unexposed early in life and exposed to famine during the fetal period, early, middle and late childhood was 2787 (23.5%), 1656 (14.0%), 2399 (20.2%), 2496 (21.0%) and 2527 (21.3%), respectively.

As shown in Table 1, there were significant differences between the fetal exposure and non-exposure groups in terms of age (*p* < 0.0001), famine severity area (*p* < 0.0001), location of residence (*p* = 0.0052), BMI (*p* = 0.0095), education level (*p* < 0.0001), drinking (*p* = 0.0276), family history of diabetes (*p* = 0.0144) and dietary pattern (*p* = 0.0003). The early childhood exposure and non-exposure groups differed significantly in age (*p* < 0.0001), famine severity area (*p* = 0.0052), area of the country (*p* = 0.0046), location of residence (*p* = 0.0460), BMI (*p* = 0.0157), education level (*p* < 0.0001) and drinking (*p* = 0.0080). The middle childhood exposure and non-exposure groups differed significantly in age (*p* < 0.0001), area of the country (*p* = 0.0009), BMI (*p* < 0.0001), education level (*p* < 0.0001), income (*p* < 0.0001), drinking (*p* = 0.0006), family history of hypertension (*p* = 0.0014) and dietary pattern (*p* = 0.0049). There were significant differences between the late-childhood-exposed and non-exposed groups in age (*p* < 0.0001), famine severity area (*p* = 0.0398), area of the country (*p* < 0.0001), location of residence (*p* = 0.0059), BMI (*p* = 0.0007), education level (*p* < 0.0001), income (*p* < 0.0001), drinking (*p* = 0.0076), physical activity (*p* = 0.0010), family history of hypertension (*p* = 0.0064) and dietary pattern (*p* = 0.0082).

### 3.2. Prevalence of MetS

As shown in Figure 1, the prevalence of MetS in the group of non-exposed and fetal, early childhood, middle childhood and late childhood exposure to famine was 41.1%, 42.7%, 41.4%, 41.0% and 45.7%, respectively. Compared with the prevalence of MetS between the non-exposed and each famine-exposed group, a significant difference was observed in the prevalence of MetS between the late-childhood-exposed group (*p* = 0.0254) and the non-exposed group, while there was no difference in MetS prevalence between the fetal period (*p* = 0.4715), early (*p* = 0.8843) and middle childhood (*p* = 0.9818)-exposed and non-exposed groups.

### 3.3. Association between Famine Exposure and MetS in Adulthood

As shown in Table 2, unadjusted for confounders, the non-exposed group was defined as the control group, the OR (95% CI) for MetS was 1.13 (1.01–1.26) in the group of early childhood exposure, 1.12 (1.00–1.25) in the group of middle childhood exposure and 1.24 (1.12–1.39) in the group of late childhood exposure.

Model I was adjusted for sex and age. Non-exposed group was defined as the control group, the OR (95% CI) for MetS was 1.27 (1.00–1.60) in the early childhood exposure group and 1.37 (1.03–1.83) in the late childhood exposure group.

Model II was further adjusted for the area of the country, location of residence, education, income, smoking, drinking, physical activity, dietary patterns, family history of hypertension or diabetes. Non-exposed group was defined as the control group, the OR (95% CI) for MetS was 1.35 (1.06–1.72) in the early childhood exposure group, 1.33 (1.02–1.73) in the middle childhood exposure group and 1.49 (1.11–2.00) in the late childhood exposure group.

Model III was further adjusted for BMI and famine severity. Non-exposed group was defined as the control group, the OR (95% CI) for MetS was 1.23 (1.00–1.51) in the fetal exposure group, 1.44 (1.11–1.87) in the early childhood exposure group, 1.50 (1.13–1.99) in the middle childhood exposure group and 1.67 (1.21–2.30) in the late childhood exposure group.

### 3.4. Stratified Analysis

As shown in Table 3, stratified analysis according to sex, BMI, famine severity, location of residence and area of the country revealed an association between early life famine exposure and risk of developing MetS in adulthood in females, overweight or obese individuals and those who had lived in severe famine areas, the city and southern China.

In females, adjusting for potential confounders, compared to the non-exposed group, famine exposure in early childhood (OR = 1.51, 95% CI = 1.08–2.12), middle childhood (OR = 1.73, 95% CI = 1.20–2.49) and late childhood (OR = 2.01, 95% CI = 1.33–3.04) increased the risk of developing MetS in adulthood. No significant association was observed in males.

In the overweight or obese population, adjusting for potential confounders, compared to the non-exposed group, exposure to famine in the fetal period (OR = 1.38, 95% CI = 1.07–1.78), early childhood (OR = 1.59, 95% CI = 1.15–2.20), middle childhood (OR = 1.79, 95% CI = 1.25–2.56) and late childhood (OR = 2.17, 95% CI = 1.45–3.24) increased the risk of developing MetS in adulthood. A similar association was not observed in non-overweight or obese populations.

In populations in areas with severe famine, adjusting for potential confounders, compared to the non-exposed group, fetal period (OR = 1.34, 95% CI = 1.01–1.77), early childhood (OR = 1.54, 95% CI = 1.08–2.21), middle childhood (OR = 1.56, 95% CI = 1.06–2.30) and late childhood (OR = 1.91, 95% CI = 1.24–2.93) exposure to famine increased the risk of developing MetS in adulthood. No significant association was observed in areas that experienced less severe famine.

In the city population, adjusting for potential confounders, compared to the non-exposed group, fetal period (OR = 1.38, 95% CI = 1.02–1.86), early childhood (OR = 1.71, 95% CI = 1.16–2.51), middle childhood (OR = 1.80, 95% CI = 1.18–2.74) and late childhood (OR = 2.04, 95% CI = 1.27–3.28) exposure to famine increased the risk of developing MetS in adulthood. No significant association was observed in the countryside population.

In the southern population, adjusting for potential confounders, compared to the non-exposed group, early childhood (OR = 1.75, 95% CI = 1.20–2.57), middle childhood (OR = 1.64, 95% CI = 1.08–2.48) and late childhood (OR = 1.93, 95% CI = 1.22–3.07) exposure to famine increased the risk of developing MetS in adulthood. No significant association was observed in the northern region.

## 4. Discussion

Our study showed that famine exposure early in life increases the risk of MetS in adulthood. Our findings were consistent with an Ethiopian famine study, which [16] found that early life famine exposure would increase the risk of MetS in adulthood, with the early life famine exposure group being three times more likely to develop MetS in adulthood than the non-exposed group. However, the Dutch [17] study found no association between early life famine exposure and the risk of developing MetS in adulthood. An explanation of the differences between famine in China and the Netherlands lies in significant differences in the duration and severity of the two famines. The Chinese famine lasted for about 3 years and covered a large part of the population [18]. The Dutch famine lasted for 6 months and therefore it did not include the entire pregnancy duration [17]. The effects of famine might depend on its duration and the organs and tissues which underwent critical periods of development during that time [19].

### 4.1. Sex Differences in Early Life Famine Exposure and MetS

This study found gender differences in the association between the famine exposure early life stage and the risk of MetS in adulthood. Famine exposure during childhood (early, middle and late childhood) increased the risk of developing MetS in adulthood in females, while it did not in males. This is consistent with previous studies by Yu, C. [12] and Wang, N. [20]. Yu, C. et al. [12] found that famine exposure early in life would increase the risk of MetS in adulthood based on the Dongfeng–Tongji cohort. Stratified analysis based on gender revealed that in females, famine exposure during childhood increased the risk of developing MetS in adulthood compared to the non-exposed group. From the 2014 Survey on Prevalence in East China for Metabolic Diseases and Risk Factors study, Wang, N. et al. reported that famine exposure during the fetal stage and childhood was significantly associated with the risk of MetS among women in adulthood [20].

Possible reasons for the different findings on the effects of early life famine exposure on MetS at the gender level are as follows: First, male fetuses were found to be more sensitive to adverse environment and had a lower chance of survival [21]. The proportion of males born during famine exposure decreases [22] and the males who survive are healthier. Second, under the influence of “son preference” and a male-dominated family structure, food is preferentially supplied to boys, resulting in the exposure of disadvantaged females to more adverse nutritional conditions. In addition, women exposed to famine early in life show a higher risk of binge eating than men [23], which may lead to an increased BMI in adulthood and thus an increased risk of MetS.

### 4.2. Different BMI of Early Life Famine Exposure and MetS

In the overweight or obese population, the risk of developing MetS in adulthood is higher when exposed to famine early in life (fetal period, early, middle and late childhood). Using data from the 2002 China National Nutrition and Health Survey (CNNHS), Li, Y. et al. [24] found that in areas affected by severe famine, compared to unexposed subjects, those exposed during the fetal stage and early childhood were at a higher risk of developing MetS in adulthood, and the association was more pronounced in subjects with Western dietary habits or overweight in adulthood. Consistent with the findings of the current study.

Research also showed similar results of the association between famine exposure and the risk of hypertension in adulthood in China. Wang, Z. et al. [25] showed that famine exposure in infancy increased the risk of hypertension in adulthood, and the association was found to be stronger in overweight and obese people according to a stratified analysis of BMI.

### 4.3. Different Famine Severity of Early Life Famine Exposure and MetS

This study used a 50% of excess mortality rate as a cut-off value to divide the severity of famine and found that exposure to severe famine early in life (fetal period, early, middle and late childhood) increased the risk of developing MetS in adulthood. This study is consistent with the research by Wang, Z. [26] and Peng, Y. [27]. Wang, Z. et al. [26] found a significant association between famine exposure during infancy (OR = 1.83) and the risk of MetS in adulthood and a stratified analysis of famine severity found a stronger association between severe famine exposure and MetS in adulthood. Peng, Y. et al. [27] analyzed 2080 study participants from the 2009 China Health and Nutrition Survey (CHNS) and found that those exposed to famine during the middle childhood and late childhood stages had a higher risk of developing MetS in adulthood. Stratified analysis found a stronger association in areas of severe famine.

### 4.4. Different Areas of Early Life Famine Exposure and MetS

Current study showed that famine exposure early in life (fetal period, early, middle and late childhood) was found to increase the risk of MetS in adulthood among the city population. Previous studies that focused on the association between famine exposure and abdominal obesity in adulthood in China also found differences between city and countryside populations. Liu, D. et al. [28] studied 35,578 participants, based on 2002 and 2010–2012 CNHS data, and found that fetal famine exposure and infant famine exposure increased the risk of abdominal obesity in adulthood. A stratified analysis found that early life famine exposure had more severe effects on those living in urban areas. The effect of early life famine exposure on MetS and abdominal obesity has different results between living areas and may be related to dietary patterns. It is suggested that people living in Chinese urban areas are more likely to have a Western dietary pattern with high energy dense [29].

In southern populations, famine exposure during childhood (early, middle and late childhood) was found to increase the risk of developing MetS in adulthood, while no similar association was observed in northern regions. This may be related to differences in dietary characteristics and lifestyle habits between the north and south of China. There is no research literature on north–south differences in the association between famine exposure in China early in life and the risk of developing MetS in adulthood.

### 4.5. Possible Mechanisms

The association between famine exposure or undernutrition early in life and the risk of developing MetS in adulthood has been verified by many studies, and the possible mechanisms are as follows: First, the fetal origin of adult diseases theory [8] suggests that when maternal malnutrition occurs, the fetus redistributes the limited nutrients to prioritize the development of vital organs, such as the brain, to ensure survival, while other organs such as the pancreas and liver undergo physiological and metabolic programmatic changes that may evolve into diseases later on during adulthood. Second, Dutch studies have found that those exposed to famine during early pregnancy are more inclined to eat a fat-intensive diet in adulthood [30]. In addition, epigenetic changes may serve as another mechanism. The results of the Dutch famine study showed that methylation levels of genes related to growth and cardiovascular metabolism (e.g., IGF2) were permanently altered after famine exposure early in life [31].

This study explores the association between famine exposure at different stages of early life and the risk of developing MetS in adulthood in conjunction with dietary patterns. The possible limitations are as follows: First, using cross-sectional survey data, this study could not infer a causal relationship between famine exposure early in life and the risk of MetS in adulthood. Second, the severity of famine in China at the time was uneven, and after the famine, residents may have migrated across regions, leading to the misclassification of the famine-exposed areas in which individuals were located.

## 5. Conclusions

Compared with the non-exposed group, the fetal, early, middle and late childhood famine-exposed groups are more likely to develop MetS in adulthood, especially the subjects who are females, overweight or obese and had lived in severe famine areas, the city and southern China.

## Figures and Tables

**Figure 1 nutrients-14-02881-f001:**
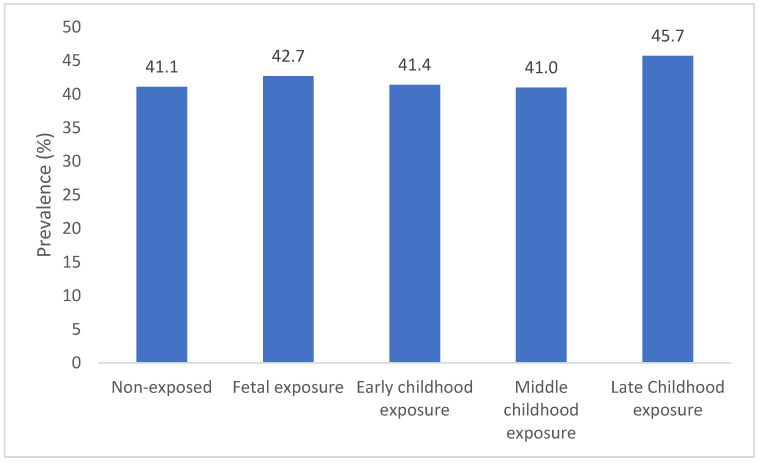
Prevalence of MetS.

**Table 1 nutrients-14-02881-t001:** General characteristic distribution in the study population.

	Non-Exposed Group	Fetal Exposure Group	Early Childhood Exposure Group	Middle Childhood Exposure Group	Late Childhood Exposure Group
Date of birth ^a^	1962–1964	1959–1961	1956–1958	1954–1956	1952–1954
N	2787	1656	2399	2496	2527
Age (years), M (IQR)	52.2 (1.0)	55.3 (1.0) *	58.2 (1.0) *	60.2 (1.0) *	62.1 (1.0) *
Male, *n* (%)	1233 (44.2)	714 (43.1)	1073 (44.7)	1126 (45.1)	1160 (45.9)
Areas with severe famine, *n* (%)	1705 (61.2)	868 (52.4) *	1376 (57.4) *	1516 (60.7)	1476 (58.4) *
South, *n* (%)	1363 (48.9)	775 (46.8)	1268 (52.9) *	1335 (53.5) *	1393 (55.1) *
City, *n* (%)	1141 (40.9)	749 (45.2) *	1048 (43.7) *	1086 (43.5)	1129 (44.7) *
BMI, *n* (%)					
<24	1148 (41.2)	748 (45.2) *	1068 (44.5) *	1181 (47.3) *	1158 (45.8) *
≥24	1639 (58.8)	908 (54.8)	1331 (55.5)	1315 (52.7)	1369 (54.2)
Education level, *n* (%)					
Low	1038 (37.2)	634 (38.3) *	1192 (49.7) *	1403 (56.2) *	1609 (63.7) *
Medium	1107 (39.7)	538 (32.5)	662 (27.6)	653 (26.2)	609 (24.1)
High	642 (23.0)	484 (29.2)	545 (22.7)	440 (17.6)	309 (12.2)
Income, *n* (%)					
Low	462 (16.6)	275 (16.6)	415 (17.3)	531 (21.3) *	549 (21.7) *
Medium	863 (31.0)	504 (30.4)	766 (31.9)	796 (31.9)	775 (30.7)
High	689 (24.7)	376 (22.7)	560 (23.3)	522 (20.9)	533 (21.1)
Very high	773 (27.7)	501 (30.3)	658 (27.4)	647 (25.9)	670 (26.5)
Smoking, *n* (%)	785 (28.2)	455 (27.5)	665 (27.7)	693 (27.8)	672 (26.6)
Drinking, *n* (%)	1087 (39.0)	591 (35.7) *	850 (35.4) *	859 (34.4) *	896 (35.5) *
Physically inactive, *n* (%)	640 (23.0)	403 (24.3)	573 (23.9)	596 (23.9)	679 (26.9) *
Family history of hypertension, *n* (%)	1052 (37.8)	662 (40.0)	876 (36.5)	837 (33.5) *	863 (34.2) *
Family history of diabetes, *n* (%)	299 (10.7)	218 (13.2) *	242 (10.1)	280 (11.2)	236 (9.3)
Dietary pattern, *n* (%)					
Egg, milk and fruit pattern	818 (29.4)	574 (34.7) *	739 (30.8)	811 (32.5) *	822 (32.5) *
Aquatic vegetable and meat pattern	980 (35.2)	507 (30.6)	874 (36.4)	779 (31.2)	797 (31.5)
Staple food soybean and nut pattern	989 (35.5)	575 (34.7)	786 (32.8)	906 (36.3)	908 (35.9)

BMI: body mass index. Non-normal distribution is expressed as M(IQR); categorical variables are expressed as n (%). a From 1 Oct. to 30 Sept. next year. * Compared to the non-exposed group, *p* < 0.05.

**Table 2 nutrients-14-02881-t002:** Association between famine exposure in early life and the risk of MetS in adulthood, OR (95% CI).

	Non-Exposed Group	Fetal Exposure Group	Early Childhood Exposure Group	Middle Childhood Exposure Group	Late Childhood Exposure Group
Unadjusted	1.00 (Ref.)	1.08 (0.95–1.22)	1.13 (1.01–1.26) *	1.12 (1.00–1.25) *	1.24 (1.12–1.39) *
Model I	1.00 (Ref.)	1.15 (0.95–1.38)	1.27 (1.00–1.60) *	1.24 (0.96–1.60)	1.37 (1.03–1.83) *
Model II	1.00 (Ref.)	1.13 (0.94–1.37)	1.35 (1.06–1.72) *	1.33 (1.02–1.73) *	1.49 (1.11–2.00) *
Model III	1.00 (Ref.)	1.23 (1.00–1.51) *	1.44 (1.11–1.87) *	1.50 (1.13–1.99) *	1.67 (1.21–2.30) *

Model I: Adjusted for sex and age. Model II: Further adjusted for area of the country, location of residence, education, income, smoking, drinking, physical activity, dietary patterns, family history of hypertension and family history of diabetes. Model III: Further adjusted for BMI and famine severity. * *p* < 0.05.

**Table 3 nutrients-14-02881-t003:** Stratified analysis of famine exposure and metabolic syndrome by sex, BMI, famine severity, location of residence and area of the country, OR (95% CI).

Stratification Factors	Non-Exposed Group	Fetal Exposure Group	Early Childhood Exposure Group	Middle Childhood Exposure Group	Late Childhood Exposure Group
Sex					
Male	1.00 (Ref.)	1.33 (0.96–1.84)	1.42 (0.94–2.14)	1.28 (0.81–2.00)	1.35 (0.82–2.23)
Female	1.00 (Ref.)	1.21 (0.93–1.56)	1.51 (1.08–2.12) *	1.73 (1.20–2.49) *	2.01 (1.33–3.04) *
BMI, kg/m²					
<24	1.00 (Ref.)	1.02 (0.72–1.46)	1.25 (0.80–1.96)	1.13 (0.69–1.83)	1.10 (0.64–1.89)
≥24	1.00 (Ref.)	1.38 (1.07–1.78) *	1.59 (1.15–2.20) *	1.79 (1.25–2.56) *	2.17 (1.45–3.24) *
Famine severity					
Less severe	1.00 (Ref.)	1.14 (0.84–1.53)	1.38 (0.94–2.02)	1.48 (0.97–2.26)	1.44 (0.89–2.33)
Serious	1.00 (Ref.)	1.34 (1.01–1.77) *	1.54 (1.08–2.21) *	1.56 (1.06–2.30) *	1.91 (1.24–2.93) *
Residence location					
City	1.00 (Ref.)	1.38 (1.02–1.86) *	1.71 (1.16–2.51) *	1.80 (1.18–2.74) *	2.04 (1.27–3.28) *
Countryside	1.00 (Ref.)	1.11 (0.84–1.47)	1.25 (0.88–1.79)	1.28 (0.87–1.88)	1.40 (0.91–2.17)
Area of the country					
South	1.00 (Ref.)	1.32 (0.98–1.77)	1.75 (1.20–2.57) *	1.64 (1.08–2.48) *	1.93 (1.22–3.07) *
North	1.00 (Ref.)	1.15 (0.87–1.52)	1.20 (0.84–1.71)	1.40 (0.94–2.07)	1.48 (0.95–2.31)

OR (95% CI) adjusted for sex, age, area of the country, location of residence, education, income, smoking, drinking, physical activity, dietary patterns, family history of hypertension, family history of diabetes and BMI, famine severity, which did not include stratification factors. * *p* < 0.05.

## Data Availability

The data are not allowed to be disclosed according to the National Institute for Nutrition and Health, Chinese Center for Disease Control and Prevention.

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
