# Peer review of "Association between Early Life Famine Exposure and Metabolic Syndrome in Adulthood"

_nutrients, 2022, doi:10.3390/nu14142881_

Round 1

Reviewer 1 Report

Yao et al. analyzed the association between early-life exposure to famine and manifestation of metabolic syndrome (MS) in adulthood in a large cohort of Chinese. It has been concluded that fetal, early childhood, middle childhood, and late childhood exposure increases to risk to develop MS, particularly if females, overweight or obese individuals, those who came from areas experiencing severe famine, from cities, and southern China.

It is a clearly written, well-planned, and conducted study, bringing important data.

I only have minor comments.

Introduction: the hypothesis is missing.

Statistical analysis: it remains unclear which data were compared using the ANOVA (why post-hoc test to localize the differences between the famine groups was not employed), and which were evaluated using the non-parametric test. In Table 1 IQ spacing is indicated, but not explained as what it actually means, and thus it is unclear whether all data were not fitting the normal distribution.

Results: section 3.3.: I would strongly suggest mentioning that the data are given in Table 2 at the beginning of the paragraph. Otherwise, it is unclear that the text refers only to significant differences (e.g., why some exposure groups are not mentioned).

Discussion: 1st paragraph: could the difference between the outcomes of the Dutch vs. Chinese famine be contributed by the fact that after the blockade of Amsterdam was broken, the food supply chain was functioning, thus starvation was followed in a short time by a relative surplus of foods? Is it possible to compare to the organization/development of food supply after Chinese famines?    

Author Response

Thank you for your detailed suggestions, we have made the corresponding changes.

We need to clarify that the last winter of the Second World War was marked by a severe famine in the Netherlands that lasted for about six months. Per capita daily rations declined from about 1,400 kcal in October 1944 to less than 1,000 kcal in late November 1944. From December 1944 to April 1945, when the Dutch famine was at its worst, the population received 400-800 kilocalories of rations per day, which was less than a quarter of the pre-famine ration. The famine ended abruptly in early May 1945 when the Netherlands was liberated and food supplies returned to normal levels.

From 1959-1961, China's grain production declined rapidly due to natural disasters and political factors. Compared with 1958, grain production fell by 15% in 1959 and grain supply fell by 30% in 1960-1961. China's famine occurred mainly because of the nationwide shortage of grain supply. after 1961, China's grain supply gradually recovered.

Combined with the above background, we believe that the large differences in the duration and severity of the famine are the main reasons for the different findings of the Chinese famine and the Dutch famine.

Thank you again for your suggestions, which have improved our article.

Reviewer 2 Report

This article analyzes the association of early life (fetal and childhood) famine exposure in China with metabolic syndrome in middle-aged adults (median age 58 y). The data were collected from 11,865 adults, including 23.5% of unexposed participants. The remaining exposed subjects were stratified into fetal, early, middle and late childhood exposure groups. The exposure was assumed if the subject lived in the area affected by famine in his/her childhood (based on excess mortality in such area at that time).

There are some problems with presentation of the results and general style of the article. In Table 1 the asterisk denoting significance is placed next to the variables in the first column on the left. It should be next to numbers significantly different from those in the non-exposed groups for each variable. Also, what is (1.0) next to each median for age? In Figure 1 the prevalence of MetS scale starts from 35%, making the differences seem artificially large.

The Abstract should be rewritten in a more concise and less repetitive style. There is no need to repeat that the associations were “stronger” for each variable and period of childhood (stronger than what?) and list each OR and 95 %CI. It is enough to point out the significant differences.

Line 15. It should be “were used”.

Line 26. It should be “areas of severe famine”.

Line 36-37. Remove “with a higher risk of MetS in adulthood”.

Line 55. It should be “to test the hypothesis in humans”.

Line 56-58.  It should be “to verify the hypothesis that the early life famine has negative effects in adulthood.”

Line 70. It should be “number”.

Line 75. It should be “long period”, not “large period”.

Line 89. It should be “in four parts”.

In the subchapter 2.5.3. each relevant definition should start on a new line.

Line 151. The first “and” should be removed.

Line 189, 226. It should be “period”.

Subchapter 3.3 is written in clipped style. The first sentence (?) has no verb. The text repeats the information found in Table 2 without referring to it. The subchapter 3.2  does not refer to Fig.1. The subchapter 3.4 again uses the word “stronger” without qualification (see above: stronger than what?).

Line 235 and 241. “Association” is better than “result”.

Line 246. It should be “is”, not “was” The subchapter 3.4 basically repeats all information available in Table 3.

Line 257-8. “Ethiopian famine study” repeated twice.

Line 260. It should be “three” not “2.94”.

Line 268 “Explanation” misspelled.

Line 263. It should be “lies in”, not “reveals”.

Line 265-6. It should be “…6 months, and therefore it did not include the entire pregnancy duration”.

 Line 274 and many below. Why are the names of researchers entirely capitalized?

Line 288. It should be “food”, not “material”.

Line 320. It should be “In our current study…”

Line 324. It should be “35,578 participants, …data,”

Line 326-7. OR alone does not denote significance.

Line 327. Remove “of living area”.

Line 330, What does it mean “differently”?

Line 332. Remove “high”.

Line 336. Remove “economic” How does weather relate to famine?

Line 342. It should be “many studies [references]”.

Line346. Remove “would”.

Line 347. It should be “may evolve”.

Line 350-1. It should be “that methylation levels of genes related to growth and… were permanently altered…”.

Line 355. Remove “However…some”. It should be “The possible limitations include…”.

Line 364-5. Remove “for those”, replace with “the subjects”. Remove “populations have high risk of MetS in adulthood”. In addition, it should be clearly stated that childhood famine may be one of the reasons for or be associated with obesity/overweight in late adulthood. Using past tense (“were overweight or obese”) implies that the girls had excessive weight already in as children.

References suffer from uneven capitalization of the titles and the names of the authors [27], as well as reporting the pages of the articles.

The manuscript should be carefully revised in preparation for possible publication.

Author Response

Thank you for your meticulous revisions, which we have made:

(1) Revised Table 1 and Figure 1.

(1) Streamlined the abstract.

(1) Carefully revised the text.

We need to clarify that in this article, age is a non-normally distributed variables and is described using the median (interquartile spacing).

Thank you again for your suggestions, which have improved our article.
